# Average-case hardness of RIP certification

**Tengyao Wang**
Centre for Mathematical Sciences
Cambridge, CB3 0WB, United Kingdom
`t.wang@statslab.cam.ac.uk`

**Quentin Berthet**
Centre for Mathematical Sciences
Cambridge, CB3 0WB, United Kingdom
`q.berthet@statslab.cam.ac.uk`

**Yaniv Plan**
1986 Mathematics Road
Vancouver BC V6T 1Z2, Canada
`yaniv@math.ubc.ca`

## Abstract

The restricted isometry property (RIP) for design matrices gives guarantees for optimal recovery in sparse linear models. It is of high interest in compressed sensing and statistical learning. This property is particularly important for computationally efficient recovery methods. As a consequence, even though it is in general NP-hard to check that RIP holds, there have been substantial efforts to find tractable proxies for it. These would allow the construction of RIP matrices and the polynomial-time verification of RIP given an arbitrary matrix. We consider the framework of average-case certifiers, that never wrongly declare that a matrix is RIP, while being often correct for random instances. While there are such functions which are tractable in a suboptimal parameter regime, we show that this is a computationally hard task in any better regime. Our results are based on a new, weaker assumption on the problem of detecting dense subgraphs.

## Introduction

In many areas of data science, high-dimensional signals contain rich structure. It is of great interest to leverage this structure to improve our ability to describe characteristics of the signal and to make future predictions. Sparsity is a structure of wide applicability (see, e.g. Mallat, 1999; Rauhut and Foucart, 2013; Eldar and Kutyniok, 2012), with a broad literature dedicated to its study in various scientific fields.

The sparse linear model takes the form $y = X\beta + \varepsilon$, where $y \in \mathbf{R}^n$ is a vector of observations, $X \in \mathbf{R}^{n \times p}$ is a *design matrix*, $\varepsilon \in \mathbf{R}^n$ is noise, and the vector $\beta \in \mathbf{R}^p$ is assumed to have a small number $k$ of non-zero entries. Estimating $\beta$ or the *mean response*, $X\beta$, are among the most widely studied problems in signal processing, as well as in statistical learning. In high-dimensional problems, one would wish to recover $\beta$ with as few observations as possible. For an incoherent design matrix, it is known that an order of $k^2$ observations suffice (Donoho, Elad and Temlyakov, 2006; Donoho and Elad, 2003). However, this appears to require a number of observations far exceeding the information content of $\beta$, which has only $k$ variables, albeit with unknown locations.

This dependence in $k$ can be greatly improved by using design matrices that are almost isometries on some low dimensional subspaces, i.e., matrices that satisfy the *restricted isometry property* with parameters $k$ and $\theta$, or RIP$(k, \theta)$ (see Definition 1.1). It is a highly robust property, and in fact implies that many different polynomial time methods, such as greedy methods (Blumensath and Davies, 2009; Needell and Tropp, 2009; Dai and Milenkovic, 2009) and convex optimization (Candès, 2008; Candès, Romberg and Tao, 2006b; Candès and Tao, 2005), are stable in recovering $\beta$.

Random matrices are known to satisfy the RIP when the number $n$ of observation is more than about $k \log(p)/\theta^2$. These results were developed in the field of *compressed sensing* (Candès, Romberg and Tao, 2006a; Donoho, 2006; Rauhut and Foucart, 2013; Eldar and Kutyniok, 2012) where the use of randomness still remains pivotal for near-optimal results. Properties related to the conditioning of design matrices have also been shown to play a key role in the statistical properties of computationally efficient estimators of $\beta$ (Zhang, Wainwright and Jordan, 2014). While the assumption of randomness allows great theoretical leaps, it leaves open questions for practitioners.

Scientists working on data closely following this model cannot always choose their design matrix $X$, or at least choose one that is completely random. Moreover, it is in general practically impossible to check that a given matrix satisfies these desired properties, as RIP certification is NP-hard (Bandeira et al., 2012). Having access to a function, or statistic, of $X$ that could be easily computed, which determines how well $\beta$ may be estimated, would therefore be of a great help. The search for such statistics has been of great importance for over a decade now, and several have been proposed (d'Aspremont and El Ghaoui, 2011; Lee and Bresler, 2008; Juditsky and Nemirovski, 2011; d'Aspremont, Bach and El Ghaoui, 2008). Perhaps the simplest and most popular is the *incoherence parameter*, which measures the maximum inner product between distinct, normalized, columns of $X$. However, all of these are known to necessarily fail to guarantee good recovery when $p \geq 2n$ unless $n$ is of order $k^2$ (d'Aspremont and El Ghaoui, 2011). Given a specific problem instance, the strong recovery guarantees of compressed sensing cannot be verified based on these statistics.

In this article, we study the problem of *average-case certification* of the Restricted Isometry Property (RIP). A certifier takes as input a design matrix $X$, always outputs 'false' when $X$ does not satisfy the property, and outputs 'true' for a large proportion of matrices (see Definition 2.1). Indeed, worst-case hardness does not preclude a problem from being solvable for most instances. The link between restricted isometry and incoherence implies that polynomial time certifiers exists in a regime where $n$ is of order $k^2 \log(p)/\theta^2$. It is natural to ask whether the RIP can be certified for sample size $n \gg k \log(p)/\theta^2$, where most matrices (with respect to, say, the Gaussian measure) are RIP. If it does, it would also provide a Las Vegas algorithm to construct RIP design matrices of optimal sizes. This should be compared with the currently existing limitations for the deterministic construction of RIP matrices.

Our main result is that certification in this sense is hard even in a near-optimal regime, assuming a new, weaker assumption on detecting dense subgraphs, related to the *Planted Clique* hypothesis.

**Theorem (Informal).** *For any $\alpha < 1$, there is no computationally efficient, average-case certifier for the class $RIP_{n,p}(k, \theta)$ uniformly over an asymptotic regime where $n \ll k^{1+\alpha}/\theta^2$.*

This suggests that even in the average case, RIP certification requires almost $k^2 \log(p)/\theta^2$ observations. This contrasts highly with the fact that a random matrix satisfies RIP with high probability when $n$ exceeds about $k \log(p)/\theta^2$. Thus, there appears to be a large gap between what a practitioner may be able to certify given a specific problem instance, and what holds for a random matrix. On the other hand, if a certifier is found which fills this gap, the result would not only have huge practical implications in compressed sensing and statistical learning, but would also disprove a long-standing conjecture from computational complexity theory.

We focus solely on the restricted isometry property, but other conditions under which compressed sensing is possible are also known. Extending our results to the restricted eigenvalue condition Bickel, Ritov and Tsybakov (2009) or other conditions (see, van de Geer and Buhlmann, 2009, and references therein) is an interesting path for future research.

Our result shares many characteristics with a hypothesis by Feige (2002) on the hardness of refuting random satisfiability formulas. Indeed, our statement is also about the hardness of verifying that a property holds for a particular instance (RIP for design matrices, instead of unsatisfiability for boolean formulas). It concerns a regime where such a property should hold with high probability ($n$ of order $k^{1+\alpha}/\theta^2$, linear regime for satisfiability), cautiously allowing only one type of errors, false negatives, for a problem that is hard in the worst case. In these two examples, such certifiers exist in a sub-optimal regime. Our problem is conceptually different from results regarding the worst-case hardness of certifying this property (see, e.g. Bandeira et al., 2012; Koiran and Zouzias, 2012; Tillmann and Pfetsch, 2014). It is closer to another line of work concerned with computational lower bounds for statistical learning problems based on average-case assumptions. The planted clique assumption has been used to prove computational hardness results for statistical problems such as estimation and testing of sparse principal components (Berthet and Rigollet, 2013a,b; Wang, Berthet

and Samworth, 2016), testing and localization of submatrix signals (Ma and Wu, 2013; Chen and Xu, 2014), community detection (Hajek, Wu and Xu, 2015) and sparse canonical correlation analysis (Gao, Ma and Zhou, 2014). The intractability of noisy parity recovery problem (Blum, Kalai and Wasserman, 2003) has also been used recently as an average-case assumption to deduce computational hardness of detection of satisfiability formulas with lightly planted solutions (Berthet and Ellenberg, 2015). Additionally, several unconditional computational hardness results are shown for statistical problems under constraints of learning models (Feldman et al., 2013). The present work has two main differences compared to previous computational lower bound results. First, in a detection setting, these lower bounds concern two specific distributions (for the null and alternative hypothesis), while ours is valid for all sub-Gaussian distributions, and there is no alternative distribution. Secondly, our result is not based on the usual assumption for the Planted Clique problem. Instead, we use a weaker assumption on a problem of detecting planted dense graphs. This does not mean that the planted graph is a random graph with edge probability $q > 1/2$ as considered in (Arias-Castro and Verzelen, 2013; Bhaskara et al., 2010; Awasthi et al., 2015), but that it can be *any graph* with an unexpectedly high number of edges (see section 3.1). This choice is made to strengthen our result: it would 'survive' the discovery of an algorithm that would use very specific properties of cliques (or even of random dense graphs) to detect their presence. As a consequence, the analysis of our reduction is more technically complicated.

Our work is organized in the following manner: We recall in Section 1 the definition of the restricted isometry property, and some of its known properties. In Section 2, we define the notion of certifier, and prove the existence of a computationally efficient certifier in a sub-optimal regime. Our main result is developed in Section 3, focused on the hardness of average-case certification. The proofs of the main results are in Appendix A of the supplementary material and those of auxiliary results in Appendix B of the the supplementary material.

# 1 Restricted Isometric Property

## 1.1 Formulation

We use the definition of Candès and Tao (2005), who introduced this notion. Below, for a vector $u \in \mathbf{R}^p$, we define $\|u\|_0$ is the number of its non-zero entries.

**Definition** (RIP). A matrix $X \in \mathbb{R}^{n \times p}$ satisfies the *restricted isometry property* with sparsity $k \in \{1, \ldots, p\}$ and distortion $\theta \in (0, 1)$, denoted by $X \in \mathrm{RIP}_{n,p}(k, \theta)$, if it holds that

$$1 - \theta \le \|Xu\|_2^2 \le 1 + \theta,$$

for every $u \in \mathbb{S}^{p-1}(k) := \{u \in \mathbb{R}^p : \|u\|_2 = 1, \|u\|_0 \le k\}$.

This can be equivalently defined by a property on submatrices of the design matrix: $X$ is in $\mathrm{RIP}_{n,p}(k, \theta)$ if and only if for any set $S$ of $k$ columns of $X$, the submatrix, $X_{*S}$, formed by taking any these columns is almost an isometry, i.e. if the spectrum of its Gram matrix is contained in the interval $[1 - \theta, 1 + \theta]$:

$$\|X_{*S}^\top X_{*S} - I_k\|_{\mathrm{op}} \le \theta\,.$$

Denote by $\| \cdot \|_{\mathrm{op},k}$ the *k-sparse operator norm*, defined for a matrix $A$ as $\|A\|_{\mathrm{op},k} = \sup_{x \in \mathbb{S}^{p-1}(k)} \|Ax\|_2$. This yields another equivalent formulation of the RIP property: $X \in \mathrm{RIP}_{n,p}(k, \theta)$ if and only if

$$\|X^\top X - I_p\|_{\mathrm{op},k} \le \theta\,.$$

We assume in the following discussion that the distortion parameter $\theta$ is upper-bounded by 1. For $v \in \mathbb{R}^p$ and $T \subseteq \{1, \ldots, p\}$, we write $v_T$ for the $\#T$-dimensional vector obtained by restricting $v$ to coordinates indexed by $T$. Similarly, for an $n \times p$ matrix $A$ and subsets $S \subseteq \{1, \ldots, n\}$ and $T \subseteq \{1, \ldots, p\}$, we write $A_{S*}$ for the submatrix obtained by restricting $A$ to rows indexed by $S$, $A_{*T}$ for the submatrix obtained by restricting $A$ to columns indexed by $T$.

## 1.2 Generation via random design

Matrices that satisfy the restricted isometry property have many interesting applications in high-dimensional statistics and compressed sensing. However, there is no known way to generate them

deterministically in general. It is even NP-hard to check whether a given matrix $X$ belongs to $\mathrm{RIP}_{n,p}(k,\theta)$ (see, e.g Bandeira et al., 2012). Several deterministic constructions of RIP matrices exist for sparsity level $k \lesssim \theta\sqrt{n}$. For example, using equitriangular tight frames and Gershgorin's circle theorem, one can construct RIP matrices with sparsity $k \leq \sqrt{n}$ and distortion $\theta$ bounded away from 0 (see, e.g. Bandeira et al., 2012). The limitation $k \leq \theta\sqrt{n}$ is known as the 'square root bottleneck'. To date, the only constructions that break the 'square root bottleneck' are due to Bourgain et al. (2011) and Bandeira, Mixon and Moreira (2014), both of which give RIP guarantee for $k$ of order $n^{1/2+\epsilon}$ for some small $\epsilon > 0$ and fixed $\theta$ (the latter construction is conditional on a number-theoretic conjecture being true).

Interestingly though, it is easy to generate large matrices satisfying the restricted isometry property through random design, and compared to the fixed design matrices mentioned in the previous paragraph, these random design constructions are much less restrictive on the sparsity level, typically allowing $k$ up to the order $n/\log(p)$ (assuming $\theta$ is bounded away from zero). They can be constructed easily from any centred sub-Gaussian distribution. We recall that a distribution $Q$ (and its associated random variable) is said to be sub-Gaussian with parameter $\sigma$ if $\int_{\mathbb{R}} e^{\lambda x}\, dQ(x) \leq e^{\lambda^2\sigma^2/2}$ for all $\lambda \in \mathbb{R}$.

**Definition.** Define $\mathcal{Q} = \mathcal{Q}_\sigma$ to be the set of sub-Gaussian distributions $Q$ over $\mathbb{R}$ with zero mean, unit variance, and sub-Gaussian parameter at most $\sigma$.

The most common choice for a $Q \in \mathcal{Q}$ is the standard normal distribution $\mathcal{N}(0,1)$. Note that by Taylor expansion, for any $Q \in \mathcal{Q}$, we necessarily have $\sigma^2 \geq \int_{\mathbb{R}} x^2\, dQ(x) = 1$. In the rest of the paper, we treat $\sigma$ as fixed. Define the normalized distribution $\tilde{Q}$ to be the distribution of $Z/\sqrt{n}$ for $Z \sim Q$. The following well-known result states that by concentration of measure, random matrices generated with distribution $\tilde{Q}^{\otimes(n\times p)}$ satisfy restricted isometries (see, e.g. Candès and Tao (2005) and Baraniuk et al. (2008)). For completeness, we include a proof that establishes these particular constants stated here. All proofs are deferred to Appendix A or Appendix B of the supplementary material.

**Proposition 1.** *Suppose $X$ is a random matrix with distribution $\tilde{Q}^{\otimes(n\times p)}$, where $Q \in \mathcal{Q}$. It holds that*

$$\mathbb{P}\big(X \in \mathrm{RIP}_{n,p}(k,\theta)\big) \geq 1 - 2\exp\left\{k\log\left(\frac{9ep}{k}\right) - \frac{n\theta^2}{256\sigma^4}\right\}. \tag{1}$$

In order to clarify the notion of asymptotic regimes used in this paper, we introduce the following definition.

**Definition.** For $0 \leq \alpha \leq 1$, define the asymptotic regime

$$\mathcal{R}_\alpha := \left\{(p_n, k_n, \theta_n)_n : p, k \to \infty \text{ and } n \gg \frac{k_n^{1+\alpha}\log(p_n)}{\theta_n^2}\right\}.$$

We note that in this notation, Proposition 1 implies that for $(p,k,\theta) = (p_n, k_n, \theta_n) \in \mathcal{R}_0$ we have, $\lim_{n\to\infty} \tilde{Q}^{\otimes(n\times p)}(X \in \mathrm{RIP}_{n,p}(k,\theta)) = 1$, and this convergence is uniform over $Q \in \mathcal{Q}$.

## 2 Certification of Restricted Isometry

### 2.1 Objectives and definition

In practice, it is useful to know with certainty whether a particular realization of a random design matrix satisfies the RIP condition. It is known that the problem of deciding if a given matrix is RIP is NP-hard (Bandeira et al., 2012). However, NP-hardness is a only a statement about worst-case instances. It would still be of great use to have an algorithm that can correctly decide RIP property for an average instance of a design matrix, with some accuracy. Such an algorithm should identify a high proportion of RIP matrices generated through random design and make no false positive claims. We call such an algorithm an *average-case certifier*, or a *certifier* for short.

**Definition** (Certifier)**.** Given a parameter sequence $(p,k,\theta) = (p_n, k_n, \theta_n)$, we define a *certifier for* $\tilde{Q}^{\otimes(n\times p)}$-*random matrices* to be a sequence $(\psi_n)_n$ of measurable functions $\psi_n : \mathbb{R}^{n\times p} \to \{0,1\}$, such that

$$\psi_n^{-1}(1) \subseteq \mathrm{RIP}_{n,p}(k,\theta) \qquad \text{and} \qquad \limsup_{n\to\infty} \tilde{Q}^{\otimes(n\times p)}\big(\psi_n^{-1}(0)\big) \leq 1/3. \tag{2}$$

Note the definition of a certifier depends on both the asymptotic parameter sequence $(p_n, k_n, \theta_n)$ and the sub-Gaussian distribution $Q$. However, when it is clear from the context, we will suppress the dependence and refer to certifiers for $\text{RIP}_{n,p}(k, \theta)$ properties of $\tilde{Q}^{\otimes(n \times p)}$-random matrices simply as 'certifiers'.

The two defining properties in (2) can be understood as follows. The first condition means that if a certifier outputs 1, we know with certainty that the matrix is RIP. The second condition means that the certifier is not overly conservative; it is allowed to output 0 for at most one third (with respect to $\tilde{Q}^{\otimes(n \times p)}$ measure) of the matrices. The choice of $1/3$ in the definition of a certifier is made to simplify proofs. However, all subsequent results will still hold if we replace $1/3$ by any constant in $(0, 1)$. In view of Proposition 1, the second condition in (2) can be equivalently stated as

$$\liminf_{n \to \infty} \tilde{Q}^{\otimes(n \times p)}\big\{\psi_n(X) = 1 \mid X \in \text{RIP}_{n,p}(k, \theta)\big\} \geq 2/3.$$

With such a certifier, given an arbitrary problem fitting the sparse linear model, the matrix $X$ could be tested for the restricted isometry property, with some expectation of a positive result. This would be particularly interesting given a certifier in the parameter regime $n \ll \theta_n^2 k_n^2$, in which presently known polynomial-time certifiers cannot give positive results.

Even though it is not the main focus of our paper, we also note that a certifier $\psi$ with the above properties for some distribution $Q \in \mathcal{Q}$ would form a certifier/distribution couple $(\psi, Q)$, that yields in the usual manner a Las Vegas algorithm to generate RIP matrices. The (random) algorithm keeps generating random matrices $X \sim \tilde{Q}^{\otimes(n \times p)}$ until $\psi_n(X) = 1$. The number of times that the certifier is invoked has a geometric distribution with success probability $\tilde{Q}^{\otimes(n \times p)}\big(\psi_n^{-1}(1)\big)$. Hence, the Las Vegas algorithm runs in randomized polynomial time if and only if $\psi_n$ runs in randomized polynomial time.

## 2.2 Certifier properties

Although our focus is on algorithmically efficient certifiers, we establish first the properties of a certifier that is computationally intractable. This certifier serves as a benchmark for the performance of other candidates. Indeed, we exhibit in the following proposition a certifier, based on the $k$-sparse operator norm, that works uniformly well in the same asymptotic parameter regime $\mathcal{R}_0$, where $\tilde{Q}^{\otimes(n \times p)}$-random matrices are RIP with asymptotic probability 1. For clarity, we stress that our criterion when judging a certifier will always be its uniform performance over asymptotic regimes $\mathcal{R}_\alpha$ for some $\alpha \in [0, 1]$.

**Proposition 2.** *Suppose* $(p, k, \theta) = (p_n, k_n, \theta_n) \in \mathcal{R}_0$. *Furthermore, Let* $Q \in \mathcal{Q}$ *and* $X \sim \tilde{Q}^{\otimes(n \times p)}$. *Then the sequence of tests* $(\psi_{op,k})_n$ *based on sparse operator norms, defined by*

$$\psi_{op,k}(X) := \mathbf{1}\left\{\|X^\top X - I_p\|_{op,k} \leq \theta\right\}.$$

*is a certifier for* $\tilde{Q}^{\otimes(n \times p)}$-*random matrices.*

By a direct reduction from the clique problem, one can show that it is NP-hard to compute the $k$-sparse operator norm of a matrix. Hence the certifier $\psi_{op,k}$ is computationally intractable. The next proposition concerns the certifier property of a test based on the maximum incoherence between columns of the design matrix. It follows directly from a well-known result on the incoherence parameter of a random matrix (see, e.g. Rauhut and Foucart (2013, Proposition 6.2)) and allows the construction of a polynomial-time certifier that works uniformly well in the asymptotic parameter regime $\mathcal{R}_1$.

**Proposition 3.** *Suppose* $(p, k, \theta) = (p_n, k_n, \theta_n)$ *satisfies* $n \geq 196\sigma^4 k^2 \log(p)/\theta^2$. *Let* $Q \in \mathcal{Q}$ *and* $X \sim \tilde{Q}^{\otimes(n \times p)}$, *then the tests* $\psi_\infty$ *defined by*

$$\psi_\infty(X) := \mathbf{1}\left\{\|X^\top X - I_p\|_\infty \leq 14\sigma^2\sqrt{\frac{\log(p)}{n}}\right\}$$

*is a certifier for* $\tilde{Q}^{\otimes(n \times p)}$-*random matrices.*

Proposition 3 shows that, when the sample size $n$ is above $k^2 \log(p)/\theta^2$ in magnitude (in particular, this is satisfied asymptotically when $(p, k, \theta) = (p_n, k_n, \theta_n) \in \mathcal{R}_1$), there is a polynomial time certifier. In other words, in this high-signal regime, the average-case decision problem for RIP property is much more tractable than indicated by the worst-case result. On the other hand, the certifier in Proposition 3 works in a much smaller parameter range when compared to $\psi_{\mathrm{op},k}$ in Proposition 2. Combining Proposition 2 and 3, we have the following schematic diagram (Figure 1). When the sample size is lower than specified in $\mathcal{R}_0$, the property does not hold, with high probability, and no certifier exists. A computationally intractable certifier works uniformly over $\mathcal{R}_0$. On the other end of the spectrum, when the sample size is large enough to be in $\mathcal{R}_1$, a simple certifier based on the maximum incoherence of the design matrix is known to work in polynomial time. This leaves open the question of whether (randomized) polynomial time certifiers can work uniformly well in $\mathcal{R}_0$, or $\mathcal{R}_\alpha$ for any $\alpha \in [0, 1)$. We will see in the next section that, assuming a weaker variant of the Planted Clique hypothesis from computational complexity theory, $\mathcal{R}_1$ is essentially the largest asymptotic regime where a randomized polynomial time certifier can exist.

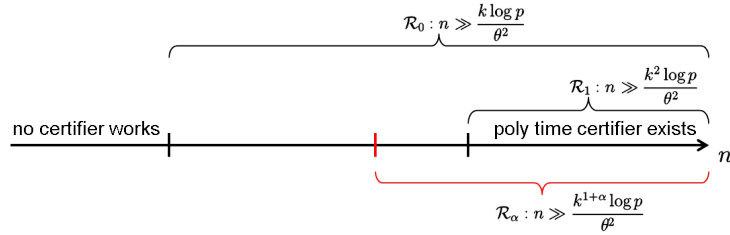

Figure 1: Schematic digram for existence of certifiers in different asymptotic regimes.

## 3  Hardness of Certification

### 3.1  Planted dense subgraph assumptions

We show in this section that certification of RIP property is an average-case hard problem in the parameter regime $\mathcal{R}_\alpha$ for any $\alpha < 1$. This is precisely the regime not covered by Proposition 3. The average-case hardness result is proved via reduction to the planted dense subgraph assumption.

For any integer $m \geq 0$, denote $\mathbb{G}_m$ the collection of all graphs on $m$ vertices. We write $V(G)$ and $E(G)$ for the set of vertices and edges of a graph $G$. For $H \in \mathbb{G}_\kappa$ where $\kappa \in \{0, \dots, m\}$, let $\mathcal{G}(m, 1/2, H)$ be the random graph model that generates a random graph $G$ on $m$ vertices as follows. It first picks $\kappa$ random vertices $K \subseteq V(G)$ and plants an isomorphic copy of $H$ on these $\kappa$ vertices, then every pair of vertices not in $K \times K$ is connected by an edge independently with probability $1/2$. We write $\mathbf{P}_H$ for the probability measure on $\mathbb{G}_m$ associated with $\mathcal{G}(m, 1/2, H)$. Note that if $H$ is the empty graph, then $\mathcal{G}(m, 1/2, \emptyset)$ describes the Erdős–Rényi random graph. With a slight abuse of notation, we write $\mathbf{P}_0$ in place of $\mathbf{P}_\emptyset$. On the other hand, for $\epsilon \in (0, 1/2]$, if $H$ belongs to the set

$$\mathcal{H} = \mathcal{H}_{\kappa,\epsilon} := \left\{ H \in \mathbb{G}_\kappa : \#E(H) \geq (1/2 + \epsilon)\frac{\kappa(\kappa - 1)}{2} \right\},$$

then $\mathcal{G}(m, 1/2, H)$ generates random graphs that contain elevated local edge density. The planted dense graph problem concerns testing apart the following two hypotheses:

$$H_0 : G \sim \mathcal{G}(m, 1/2, \emptyset) \qquad \text{and} \qquad H_1 : G \sim \mathcal{G}(m, 1/2, H) \text{ for some } H \in \mathcal{H}_{\kappa,\epsilon}. \qquad (3)$$

It is widely believed that for $\kappa = O(m^{1/2-\delta})$, there does not exist randomized polynomial time tests to distinguish between $H_0$ and $H_1$ (see, e.g. Jerrum (1992); Feige and Krauthgamer (2003); Feldman et al. (2013)). More precisely, we have the following assumption.

**Assumption (A1) 1.** Fix $\epsilon \in (0, 1/2]$ and $\delta \in (0, 1/2)$. let $(\kappa_m)_m$ be any sequence of integers such that $\kappa_m \to \infty$ and $\kappa_m = O(m^{1/2-\delta})$. For any sequence of randomized polynomial time tests $(\phi_m : \mathbb{G}_m \to \{0, 1\})_m$, we have

$$\liminf_m \left\{ \mathbf{P}_0\big(\phi(G) = 1\big) + \max_{H \in \mathcal{H}_{\kappa,\epsilon}} \mathbf{P}_H\big(\phi(G) = 0)\big) \right\} > 1/3 \,.$$

We remark that if $\epsilon = 1/2$, then $\mathcal{H}_{\kappa,\epsilon}$ contains only the $\kappa$-complete graph and the testing problem becomes the well-known planted clique problem (cf. Jerrum (1992) and references in Berthet and Rigollet (2013a,b)).

The difficulty of this problem has been used as a primitive for the hardness of other tasks, such as cryptographic applications, in Juels and Peinado (2000), testing for $k$-wise dependence in Alon et al. (2007), approximating Nash equilibria in Hazan and Krauthgamer (2011). In this case, Assumption (A1) is a version of the planted clique hypothesis (see, e.g. Berthet and Rigollet (2013b, Assumption $\mathbf{A}_{\mathrm{PC}}$)). We emphasize that Assumption A1 is significantly milder than the planted clique hypothesis (since it allows any $\epsilon \in (0, 1/2]$), or that a hypothesis on planted random graphs. We also note that when $\kappa \geq C_\epsilon \sqrt{m}$, spectral methods can be used to detect such graphs with high probability. Indeed, when $G$ contains a graph of $\mathcal{H}$, denoting $A_G$ its adjacency matrix, then $A_G - \mathbf{1}\mathbf{1}^\top/2$ has a leading eigenvalue greater than $\epsilon(\kappa - 1)$, whereas it is of order $\sqrt{m}$ for a usual Erdős–Rényi random graph.

The following theorem relates the hardness of the planted dense subgraph testing problem to the hardness of certifying restricted isometry of random matrices. We recall that the distribution of $X$ is that of an $n \times p$ random matrix with entries independently and identically sampled from $\tilde{Q} \overset{d}{=} Q/\sqrt{n}$, for some $Q \in \mathcal{Q}$. We also write $\Psi_{\mathrm{rp}}$ for the class of randomized polynomial time certifiers.

**Theorem 4.** *Assume (A1) and fix any $\alpha \in [0, 1)$. Then there exists a sequence $(p, k, \theta) = (p_n, k_n, \theta_n) \in \mathcal{R}_\alpha$, such that there is no certifier/distribution couple $(\psi, Q) \in \Psi_{\mathrm{rp}} \times \mathcal{Q}$ with respect to this sequence of parameters.*

Our proof of Theorem 4 relies on the following ideas: Given a graph $G$, an instance of the planted clique problem in the assumed hard regime, we construct $n$ random vectors based on the adjacency matrix of a bipartite subgraph of $G$, between two random sets of vertices. Each coefficient of these vectors is then randomly drawn from one of two carefully chosen distributions, conditionally on the presence or absence of a particular edge. This construction ensures that if the graph is an Erdős–Rényi random graph (i.e. with no planted graph), the vectors are independent with independent coefficients, with distribution $\tilde{Q}$. Otherwise, we show that with high probability, the presence of an unusually dense subgraph will make it very likely that the matrix does not satisfy the restricted isometry property, for a set of parameters in $\mathcal{R}_\alpha$. As a consequence, if there existed a certifier/distribution couple $(\psi, Q) \in \Psi_{\mathrm{rp}} \times \mathcal{Q}$ in this range of parameters, it could be used - by using as input in the certifier the newly constructed matrix - to determine with high probability the distribution of $G$, violating our assumption (A1).

We remark that this result holds for *any* distribution in $\mathcal{Q}$, in contrast to computational lower bounds in statistical learning problems, that apply to a specific distribution. For the sake of simplicity, we have kept the coefficients of $X$ identically distributed, but our analysis is not dependent on that fact, and our result can be directly extended to the case where the coefficients are independent, with different distributions in $\mathcal{Q}$.

Theorem 4 may be viewed as providing an asymptotic lower bound of the sample size $n$ for the existence of a computationally feasible certifier. It establishes this computational lower bound by exhibiting some specific 'hard' sequences of parameters inside $\mathcal{R}_\alpha$, and show that any algorithm violating the computational lower bound could be exploited to solve the planted dense subgraph problem. All hardness results, whether in a worst-case (NP-hardness, or other) or the average-case (by reduction from a hard problem), are by nature statements on the impossibility of accomplishing a task in a computationally efficient manner, uniformly over a range of parameters. They are therefore always based on the construction of a 'hard' sequence of parameters used in the reduction, for which a contradiction is shown. Here, the 'hard' sequence is explicitly constructed in the proof to be some $(p, k, \theta) = (p_n, k_n, \theta_n)$ satisfying $p \geq n$ and $n^{1/(3-\alpha-4\beta)} \ll k \ll n^{1/(2-\beta)-\delta}$, for $\beta \in [0, (1-\alpha)/3)$ and any small $\delta > 0$. The tuning parameter $\beta$ is to allow additional flexibility in choosing these 'hard' sequences. More precisely, using an averaging trick first seen in Ma and Wu (2013), we are able to show that the existence of such 'hard' sequences is not confined only in the sparsity regime $k \ll n^{1/2}$. We note that in all our 'hard' sequences, $\theta_n$ must depend on $n$. An interesting extension is to see if similar computational lower bounds hold when restricted to a subset of $\mathcal{R}_\alpha$ where $\theta$ is constant.

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
