[Supplementary Material]

# Supplementary material to "Average-case hardness of RIP certification"

**Tengyao Wang**
Centre for Mathematical Sciences
Cambridge, CB3 0WB, United Kingdom
t.wang@statslab.cam.ac.uk

**Quentin Berthet**
Centre for Mathematical Sciences
Cambridge, CB3 0WB, United Kingdom
q.berthet@statslab.cam.ac.uk

**Yaniv Plan**
1986 Mathematics Road
Vancouver BC V6T 1Z2, Canada
yaniv@math.ubc.ca

## A   Proofs of Main Results

*Proof of Theorem 4.* We prove by contradiction. Assume the contrary, that $(\psi_n)_n$ is a polynomial time computable certifier for $\tilde{Q}^{\otimes(n\times p)}$-random matrices.

For $\alpha < 1$ and $0 \le \beta < \frac{1}{3}(1-\alpha)$, let $(p,k,\theta) = (p_n, k_n, \theta_n) \in \mathcal{R}_\alpha$ be a sequence satisfying $p \ge n$, $\log p = O(\log n)$, $n^{\frac{1}{3-\alpha-4\beta}} \ll k \ll n^{\frac{1}{2-\beta}-\delta}$ for some $\delta > 0$ and $\theta = \sqrt{k^{1+\alpha}\log(p)/n}$. Let $L = 10$ and $\ell = \lfloor k^\beta \rfloor$. Define $m = L\ell n$ and $\kappa = Lk$. We check that

$$\kappa^2 \asymp k^{2-\beta}k^\beta \ll n^{1-\delta}\ell \ll m^{1-\delta'}$$

for some positive $\delta'$ that depends on $\delta$ only. We remark that the purpose of introducing the extra parameter $\beta$ in the proof is mainly to show the ubiquity of parameter sequences $(p,k,\theta)$ that arrive at a contradiction. In particular, we can use positive $\beta$ values to construct sequences where $k \gg n^{1/2}$. For a first reading, it suffices to take $\beta = 0$ (i.e. $\ell = 1$), which already constitutes a proof of the theorem. When $\beta > 0$, the proof requires the additional assumption that there exists $\check{Q}$ such that for $Y_1, \dots, Y_\ell^2 \overset{\text{i.i.d.}}{\sim} \check{Q}$, $\ell^{-1}\sum_{i=1}^{\ell^2} Y_i \sim \tilde{Q}$. Note when $\beta = 0$, we can simply take $\check{Q} = \tilde{Q}$. Let $\xi$ denote the median of $\check{Q}$. By definition of the median, there exists a unique decomposition of the probability measure $\check{Q}$ as $\check{Q} = \frac{1}{2}\check{Q}^+ + \frac{1}{2}\check{Q}^-$, where $\check{Q}^+$ and $\check{Q}^-$ are probability measures supported on $(-\infty, \xi]$ and $[\xi, \infty)$ respectively.

We prove below that Algorithm 1, which runs in randomized polynomial time, can distinguish between $\mathbf{P}_0$ and $\mathbf{P}_H$ with zero asymptotic error for any choice of $H \in \mathcal{H}_{\kappa,\epsilon}$.

First, assume $G \sim \mathbf{P}_0$. Then matrix $A$ from Step 1 of Algorithm 1 have independent Rademacher entries, which implies that $X \sim \tilde{Q}^{\otimes(n\times p)}$. Therefore, by (2) in Section 2 we must have

$$\liminf \mathbf{P}_0(\phi(G) = 1) = \tilde{Q}^{\otimes(n\times p)}(\psi_n^{-1}(0)) < 1/3.$$

Next, assume $G$ is generated with probability measure $\mathbf{P}_H$ for some $H \in \mathcal{H}_{\kappa,\epsilon}$. We claim that

$$\tilde{X} \notin \mathrm{RIP}_{n,n}\left(k, \frac{ck^2}{n\ell^2}\right) \tag{1}$$

for some absolute positive constant $c$. Since

$$\frac{k^2}{n\ell^2} \gg \sqrt{\frac{k^{1+\alpha}}{n}} \gg \theta,$$

**Algorithm 1:** Pseudo-code for an algorithm to distinguish between $\mathbf{P}_0$ and $\mathbf{P}_H$.

**Input**: $m \in \mathbb{N}$, $\kappa \in \{1, \ldots, m\}$, $G \in \mathbb{G}_m$, $L \in \mathbb{N}$

**begin**

    **Step 1:** Let $N \leftarrow \lfloor m/L \rfloor$, $k \leftarrow \lfloor \kappa/L \rfloor$, $\ell \leftarrow \lfloor k^\beta \rfloor$, $n \leftarrow \lfloor N/\ell \rfloor$, $p \leftarrow p_n$. Draw $u_1, \ldots, u_N, w_1, \ldots, w_N$ uniformly at random without replacement from $V(G)$. Form $A = (A_{ij}) \in \mathbb{R}^{N \times N}$ where $A_{ij} = 2 \cdot \mathbb{1}_{\{u_i \sim w_j\}} - 1$.

    **Step 2:** Let $Y^+ = (Y_{ij}^+)$ and $Y^- = (Y_{ij}^-)$ be $N$-by-$N$ random matrices independent from all other random variables and from each other, and such that $Y_{ij}^+ \overset{\text{i.i.d.}}{\sim} \breve{Q}^+$ and $Y_{ij}^- \overset{\text{i.i.d.}}{\sim} \breve{Q}^-$. Define $Z = (Z_{ij})$ by $Z_{ij} = \mathbb{1}\{A_{ij} = 1\}Y_{ij}^+ + \mathbb{1}\{A_{ij} = -1\}Y_{ij}^-$.

    **Step 3:** For $0 \leq a, b \leq \ell - 1$, define $Z^{(a,b)} \in \mathbb{R}^{n \times n}$ by $Z_{i,j}^{(a,b)} = Z_{an+i, bn+j}$. Define $\tilde{X} \leftarrow \ell^{-1} \sum_{0 \leq a,b < \ell} Z^{(a,b)}$. Finally, let $X \leftarrow \begin{pmatrix} \tilde{X} & \tilde{X}' \end{pmatrix}$ where $\tilde{X}' \in \mathbb{R}^{n \times (p-n)}$ has entries independently drawn from distribution $\tilde{Q}$.

    **Step 4:** Let $\phi(G) \leftarrow 1 - \psi_n(X)$.

**end**

**Output**: $\phi(G)$

---

we have that for large $n$, $\tilde{X} \notin \text{RIP}_{n,n}(k, \theta)$. Hence $X$ is *a fortiori* not an $\text{RIP}_{n,p}(k, \theta)$ matrix. As a result,

$$\liminf_m \max_{H \in \mathcal{H}_{\kappa,\epsilon}} \mathbf{P}_H\big(\phi(G) = 0\big) < 1/3,$$

contradicting Assumption (**A1**).

It remains to verify the claimed result in (1). Let $K \subseteq V(G)$ be the $\kappa$-subset of vertices on which the subgraph $H$ is planted. We write $U = \{u_1, \ldots, u_N\}$ and $W = \{w_1, \ldots, w_N\}$ for the two random subsets of vertices. Let $N_{U,W;K}$ be the random variable counting the number of edges in $G$ with two endpoints in $U \cap K$ and $W \cap K$ respectively. Then

$$N_{U,W;K} = \#\Big\{\{u, w\} \in E(G) : u \in U \cap K, w \in W \cap K\Big\}$$
$$= \sum_{u \in K} \sum_{w \in K} \mathbb{1}\{u \in U\}\mathbb{1}\{w \in W\}\mathbb{1}\{u \sim w\}.$$

Define

$$\Omega_1 := \left\{ N_{U,W;K} \geq \left(\frac{1}{2} + \frac{\epsilon}{4}\right)k^2 \right\} \cap \left\{ |\#U \cap K - k| \leq \frac{\epsilon}{8}k \right\} \cap \left\{ |\#W \cap K - k| \leq \frac{\epsilon}{8}k \right\}.$$

Lemma 1 below shows that $\Omega_1$ has asymptotic probability 1. Note $\Omega_1$ is in the $\sigma$-algebra of $(U, W)$. Let $U = U_0$ and $W = W_0$ be any realization satisfying $\Omega_1$. We write $\mathbb{P}^{U_0, W_0}$ and $\mathbb{E}^{U_0, \dot{W}_0}$ as shorthand for the probability and expectation conditional on $U = U_0$ and $W = W_0$.

For each $j \in \{1, \ldots, n\}$, define $s_j := \sum_{u_i \in U \cap K} A_{i,j}$. Write $k_1 := (1 - \epsilon/8)k$ and $k_2 = (1 + \epsilon/8)k$. Let $S := \{i : u_i \in U \cap K\}$, and let $T$ be a subset of $k_1$ indices in $\{1, \ldots, n\}$ corresponding to the $k_1$ largest values of $s_j$ (breaking ties arbitrarily). Note that $S$ and $T$ are functions of $U$ and $V$. On the event $U = U_0$ and $W = W_0$, both $\#S = \#U \cap K$ and $\#W \cap K$ are bounded in the interval $[k_1, k_2]$, so in particular $k_1 \leq \#W \cap K$. We have

$$\sum_{w_j \in W \cap K} s_j = 2N_{U,W;K} - \#(U \cap K) \times \#(W \cap K) \geq \big\{(1 + \epsilon/2) - (1 + \epsilon/8)^2\big\}k^2 \geq \frac{\epsilon}{5}k^2.$$

As elements of $T$ index columns of $A$ corresponding to largest values of $s_j$s, we have that on event $\{U = U_0, W = W_0\}$,

$$\sum_{j \in T} s_j \geq \frac{\#T}{\#W \cap K}\frac{\epsilon}{5}k^2 \geq \frac{\epsilon}{5}\frac{k^2 k_1}{k_2} \geq \frac{\epsilon}{6}kk_1. \tag{2}$$

Define the unit vector $v \in \mathbb{R}^n$ by $v_T = k_1^{-1/2}\mathbf{1}_{k_1}$ and $v_{T^c} = 0$. Note that $v$ is $k_1$-sparse and hence also $k$-sparse. Conditional on $U = U_0$ and $W = W_0$, $Z_{ij} = Y_{ij}^+$ if $A_{ij} = 1$ and $Z_{ij} = Y_{ij}^-$ if $A_{ij} = -1$. By definition of $\tilde{Q}^+$ and $\tilde{Q}^-$, and the fact that $\tilde{Q}$ is not a point mass, we have $\mathbb{E}Y_{ij}^+ = -\mathbb{E}Y_{ij}^- = c_1/\sqrt{n}$ for some absolute constant $c_1 > 0$. By (2), the sum $\sum_{i \in S, j \in T} Z_{ij}$ can be bounded below in conditional expectation by

$$
\begin{aligned}
\mathbb{E}^{U_0, W_0} \sum_{i \in S, j \in T} Z_{ij} &\geq \mathbb{E}^{U_0, W_0}\left( \sum_{i \in S, j \in T} \left(\mathbb{1}\{A_{ij} = 1\}Y_{ij}^+ + \mathbb{1}\{A_{ij} = -1\}Y_{ij}^-\right) \right) \\
&= \frac{c_1}{\sqrt{n}}\left( \sum_{j \in T} s_j \right) \geq \frac{c_1}{\sqrt{n}}\frac{\epsilon}{6}kk_1 \,.
\end{aligned}
$$

By Lemma 3, both $Y_{ij}^+ - \mathbb{E}Y_{ij}^+$ and $Y_{ij}^- - \mathbb{E}Y_{ij}^-$ are sub-Gaussian with parameter at most $c_2\sigma/\sqrt{n}$ for some absolute constant $c_2 > 0$. By Hoeffding's inequality for sums of sub-Gaussian random variables (see e.g. Vershynin (2012, Proposition 5.10)),

$$
\mathbb{P}^{U_0, W_0}\left( \sum_{i \in S, j \in T} Z_{ij} > \frac{c_1\epsilon}{12\sqrt{n}}kk_1 \right) \geq 1 - 2\exp\left\{ -\frac{(\frac{c_1\epsilon}{12\sqrt{n}}kk_1)^2}{2c_2^2\sigma^2 k_1 k_2/n} \right\} \to 1. \tag{3}
$$

By (3) and the fact that $\mathbb{P}(\Omega_1) \to 1$, the event

$$
\Omega_2 := \left\{ \sum_{i \in S, j \in T} Z_{ij} \geq \frac{c_1\epsilon kk_1}{12\sqrt{n}} \right\}
$$

has asymptotic probability 1.

Now define

$$
\tilde{S} = \{i \in \{1, \ldots, n\} : u_{an+i} \in U \cap K \text{ for some } 0 \leq a \leq \ell - 1\}
$$
$$
\tilde{T} = \{j \in \{1, \ldots, n\} : w_{bn+j} \in W \cap K \text{ for some } 0 \leq b \leq \ell - 1\}
$$

Also, define $v^{(b)} = (v_{bn+1}, \ldots, v_{bn+n})^\top$ for $0 \leq b \leq \ell - 1$, $\tilde{v}_{\text{sum}} = \sum_{0 \leq b \leq \ell-1} v^{(b)}$ and $\tilde{v} = \tilde{v}_{\text{sum}}/\|\tilde{v}_{\text{sum}}\|_2$. By Lemma 6, we have $\|\tilde{v}_{\text{sum}}\|_\infty \leq c_2 k_1^{-1/2}$ with asymptotic probability 1 for some $c_2$ depending on $\beta$ only. Hence $\|\tilde{v}_{\text{sum}}\|_2 \leq c_2$. Thus, by Cauchy–Schwarz inequality, we have with asymptotic probability 1,

$$
\|\tilde{X}_{\tilde{S}*}\tilde{v}\|_2 \geq \|\tilde{v}_{\text{sum}}\|_2^{-1}(\#\tilde{S})^{-1/2}\|\tilde{X}_{\tilde{S}*}\tilde{v}_{\text{sum}}\|_1
$$

Since

$$
\tilde{X}_{\tilde{S}*}\tilde{v}_{\text{sum}} = \ell^{-1}\left( \sum_{0 \leq a,b < \ell} Z_{S*}^{(a,b)} \right)\left( \sum_{0 \leq b' < \ell} v^{(b)} \right) = \ell^{-1}\sum_{0 \leq a,b < \ell} Z_{S*}^{(a,b)}v^{(b)} + \ell^{-1}\sum_{\substack{0 \leq a,b,b' < \ell \\ b \neq b'}} Z_{S*}^{(a,b)}v^{(b')}
$$

We can bound $\|\tilde{X}_{\tilde{S}*}\tilde{v}_{\text{sum}}\|_1$ from below by the entrywise sums of the two terms above. The entrywise sum of the first term can be rewritten as $\ell^{-1}\sum_{i \in S, j \in T} Z_{ij}$, which by (3) is bounded from below by $\frac{c_3\epsilon k}{\ell\sqrt{n}}$ with asymptotic probability 1. The second term has entries with nonnegative means, hence another application of the Hoeffding's inequality shows that its contribution will be of smaller order than the first term with high probability. To summarise, we have that

$$
\|\tilde{X}_{\tilde{S}*}\tilde{v}\|_2 \geq \frac{c_3\epsilon k}{\ell\sqrt{n}}.
$$

with asymptotic probability 1. On the other hand, the submatrix $\tilde{X}_{\tilde{S}^c*}$ has independent and identically distributed entries. By Vershynin (2012, Lemma 5.9), for $i \in \tilde{S}^c$ and $1 \leq j \leq n$, $\tilde{X}_{ij} = \ell^{-1}\sum_{a,b=0}^{\ell-1} Z_{an+i,bn+j}^{(a,b)}$ is a centred sub-Gaussian random variable with sub-Gaussian parameter $\sigma/\sqrt{n}$ and variance $1/n$. Let $\tilde{X}_i$ denote the $i$th row vector of the matrix $\tilde{X}$, then conditional on $\tilde{T}$, we have that $\tilde{X}_i^\top\tilde{v}$ is also a centred sub-Gaussian random variable with parameter $\sigma/\sqrt{n}$ and variance $1/n$. Using Lemma 5, we have

$$
\mathbb{P}\left( \|\tilde{X}_{S^c*}\tilde{v}\|_2^2 - \frac{n - \#\tilde{S}}{n} \leq -\sqrt{\frac{\log n}{n - \#\tilde{S}}} \right) \leq \exp\left\{ -\frac{\log n}{64\sigma^4} \right\} \to 0.
$$

Since $\#\tilde{S} \leq k_2$ with asymptotic probability 1, the event

$$\Omega_3 := \left\{ \|\tilde{X}_{\tilde{S}^c*}\tilde{v}\|_2^2 \geq 1 - \frac{k_2}{n} - \sqrt{\frac{2\log n}{n}} \right\}$$

has asymptotic probability 1. Finally, since $\tilde{X}\tilde{v} = (\tilde{X}_{\tilde{S}*}\tilde{v}, \tilde{X}_{\tilde{S}^c*}v)^\top$, on $\Omega_2 \cap \Omega_3$,

$$\|\tilde{X}\tilde{v}\|_2^2 = \|\tilde{X}_{\tilde{S}*}\tilde{v}\|_2^2 + \|\tilde{X}_{\tilde{S}^c*}v\|_2^2 \geq 1 + \frac{c_3^2\epsilon^2 k^2}{\ell^2 n} - \frac{k_2}{n} - \sqrt{\frac{2\log n}{n}}.$$

The right hand side is at least $1 + ck^2/(n\ell^2)$ for some absolute positive constant $c$ for all large values of $n$. This verifies (1) and concludes the proof. $\qquad\square$

**Lemma 1.** *Let $G$ be a graph on $m$ vertices and $K$ a $\kappa$-subset of $V(G)$, such that the edge density of $G$ restricted to $K$ is at least $1/2 + \epsilon$. Let $n, p$ be integers less than $m/2$. Choose $u_1, \ldots, u_n$ and $w_1, \ldots, w_p$ independently at random without replacement from $V(G)$. Denote $U = \{u_1, \ldots, u_n\}$ and $W = \{w_1, \ldots, w_p\}$. Define $N_{U,W;K}$ to be the number of edges with two endpoints in $U$ and $W$ respectively. Then for $m, n, p, \kappa$ sufficiently large.*

$$\mathbb{P}\left\{ \left| \#U \cap K - \frac{n\kappa}{m} \right| \geq \frac{\epsilon}{8}\frac{n\kappa}{m} \right\} \leq \frac{64m}{\epsilon^2 n\kappa},$$

$$\mathbb{P}\left\{ \left| \#W \cap K - \frac{p\kappa}{m} \right| \geq \frac{\epsilon}{8}\frac{p\kappa}{m} \right\} \leq \frac{64m}{\epsilon^2 p\kappa},$$

$$\mathbb{P}\left\{ N_{U,W;K} \leq \left(\frac{1}{2} + \frac{\epsilon}{4}\right)\frac{np\kappa^2}{m^2} \right\} \leq \frac{16m(p\kappa + n\kappa + m)}{\epsilon^2 np\kappa^2}.$$

*Proof.* The cardinality of $U \cap K$ has $\mathrm{HyperGeom}(m, \kappa, n)$ distribution. Hence

$$\mathbb{E}(\#U \cap K) = \frac{n\kappa}{m} \quad \text{and} \quad \mathrm{var}(\#U \cap K) = n\frac{\kappa}{m}\frac{m - \kappa}{m}\frac{m - n}{m - 1} \leq \frac{n\kappa}{m}.$$

The first inequality in the lemma now follows from an application of Chebyshev's inequality. A similar argument establishes the second inequality. For the final inequality in the lemma, we have that for $\kappa$ sufficiently large,

$$\mathbb{E}(N_{U,W;K}) = \sum_{u \in K}\sum_{w \in K} \mathbb{P}(u \in U, w \in W)\mathbb{1}\{u \sim w\}$$

$$= \frac{np}{m(m-1)}\sum_{u \in K}\sum_{w \in K}\mathbb{1}\{u \sim w\} \geq \left(\frac{1}{2} + \epsilon\right)\frac{np\kappa(\kappa - 1)}{m(m-1)} \geq \left(\frac{1}{2} + \frac{\epsilon}{2}\right)\frac{np\kappa^2}{m^2}.$$

We then compute the variance of $N_{U,W;K}$ by

$$\mathrm{var}(N_{U,W;K}) = \mathrm{cov}\left( \sum_{u \in K}\sum_{w \in K}\mathbb{1}\{u \in U, w \in W, u \sim w\}, \sum_{u' \in K}\sum_{w' \in K}\mathbb{1}\{u' \in U, w' \in W, u' \sim w'\} \right)$$

$$= \sum_{u,w,u',w' \in K} \mathrm{cov}\big(\mathbb{1}\{u \in U, w \in W, u \sim w\}, \mathbb{1}\{u' \in U, w' \in W, u' \sim w'\}\big)$$

$$=: \mathrm{I} + \mathrm{II} + \mathrm{III} + \mathrm{IV},$$

where the four terms I, II, III and IV handle sums over subsets of indices $\{(u, w, u', w') \in K^4 : u \neq u', w \neq w'\}$, $\{(u, w, u', w') \in K^4 : u = u', w \neq w'\}$, $\{(u, w, u', w') \in K^4 : u \neq u', w = w'\}$ and $\{(u, w, u', w') \in K^4 : u = u', w = w'\}$ respectively.

We bound the four terms separately. For the first term, we have

$$\mathrm{I} = \sum_{u,u',w,w' \text{ distinct}} \left\{ \mathbb{P}(u, u' \in U, w, w' \in W) - \mathbb{P}(u \in U, w \in W)\mathbb{P}(u' \in U, w' \in W) \right\}\mathbb{1}\{v \sim w\}\mathbb{1}\{u' \sim w'\}$$

$$= \sum_{u,u',w,w' \text{ distinct}} \left\{ \frac{n(n-1)p(p-1)}{m(m-1)(m-2)(m-3)} - \left(\frac{np}{m(m-1)}\right)^2 \right\}\mathbb{1}\{u \sim w\}\mathbb{1}\{u' \sim w'\}.$$

When $m > \max(2n, 2p)$, the term in bracket above is non-positive, hence $\text{I} \leq 0$. For the second term, we get that

$$\text{II} = \sum_{u,w,w' \text{ distinct}} \left\{ \mathbb{P}(u \in U, w, w' \in W) - \mathbb{P}(u \in U, w \in W) \mathbb{P}(u \in U, w' \in W) \right\} \mathbb{1}\{u \sim w\} \mathbb{1}\{u' \sim w'\}$$

$$= \sum_{u,w,w' \text{ distinct}} \left\{ \frac{np(p-1)}{m(m-1)(m-2)} - \left( \frac{np}{m(m-1)} \right)^2 \right\} \mathbb{1}\{u \sim w\} \mathbb{1}\{u \sim w'\}$$

$$\leq \frac{np(p-1)}{m(m-1)(m-2)} \sum_{u,w,w' \text{ distinct}} \mathbb{1}\{u \sim w\} \mathbb{1}\{u \sim w'\} \leq \frac{np^2\kappa^3}{m^3}.$$

Similarly, we have

$$\text{III} \leq \frac{n(n-1)p\kappa(\kappa-1)(\kappa-2)}{m(m-1)(m-2)} \leq \frac{n^2 p\kappa^3}{m^3}.$$

And finally,

$$\text{IV} = \sum_{u,w \text{ distinct}} \left\{ \mathbb{P}(u \in U, w \in W) - \mathbb{P}(u \in U, w \in W)^2 \right\} \mathbb{1}\{u \sim w\} \leq \frac{np\kappa(\kappa-1)}{m(m-1)} \leq \frac{np\kappa^2}{m^2}.$$

Sum up the four terms, we get that

$$\text{var}(N_{U,W;K}) \leq \frac{np\kappa^2}{m^2} \left( \frac{p\kappa}{m} + \frac{n\kappa}{m} + 1 \right).$$

By Chebyshev's inequality, we get that

$$\mathbb{P}\left\{ N_{U,W;K} \leq \left( \frac{1}{2} + \frac{\epsilon}{4} \right) \frac{np\kappa^2}{m^2} \right\} \leq \frac{16m(p\kappa + n\kappa + m)}{\epsilon^2 np\kappa^2},$$

as desired. $\qquad\square$

## B   Auxiliary Results

*Proof of Proposition 1.* Let $X_i$ denote the $i$th row vector of $X$. Then for any fixed $u \in \mathbb{S}^p(k)$,

$$\mathbb{E}e^{\lambda(X_i^\top u)} = \prod_{1 \leq j \leq p} \mathbb{E}e^{\lambda X_{ij} u_j} \leq \prod_j e^{\lambda^2 u_j^2/(2\sigma^2 n)} = e^{\lambda^2/(2\sigma^2 n)}.$$

Apply Lemma 5 to $\|Xu\|_2^2 - 1 = n^{-1} \sum_{i=1}^n \left\{ (\sqrt{n}X_i^\top u)^2 - \mathbb{E}(\sqrt{n}X_i^\top u)^2 \right\}$, and use the fact that $\theta/(8\sigma^2) \leq 1$, we have

$$\mathbb{P}\big( 1 - \theta \leq \|Xu\|_2^2 \leq 1 + \theta \big) \geq 1 - 2e^{-n\theta^2/(64\sigma^4)}.$$

We claim that there is a set $\mathcal{N}$ of cardinality at most $\binom{p}{k} 9^k$ such that

$$\sup_{u \in \mathbb{S}^p(k)} \left| \|Xu\|_2^2 - 1 \right| \leq 2 \sup_{u \in \mathcal{N}} \left| \|Xu\|_2^2 - 1 \right| \tag{4}$$

Given (4), by union bound, we have

$$\mathbb{P}(X \in \text{RIP}(k, \theta)) = \mathbb{P}\Big( \sup_{u \in \mathbb{S}^p(k)} \left| \|Xu\|_2^2 - 1 \right| \leq \theta \Big) \geq \mathbb{P}\Big( \sup_{u \in \mathcal{N}} \left| \|Xu\|_2^2 - 1 \right| \leq \theta/2 \Big)$$

$$\geq 1 - 2\binom{p}{k} 9^k e^{-n\theta^2/(256\sigma^4)} \geq 1 - 2\exp\left\{ k\log\left( \frac{9ep}{k} \right) - \frac{n\theta^2}{256\sigma^4} \right\},$$

as desired. It remains to verify Claim (4). For any cardinality $k$ subset $J \subseteq \{1, \ldots, p\}$, let $B_J = \{u \in \mathbb{S}^p(k) : u_{J^c} = 0\}$. Each $B_J$ contains a $1/4$-net, $\mathcal{N}_J$, of cardinality at most $9^k$ (Vershynin, 2012, Lemma 5.2). Then $\mathcal{N} := \cup_J \mathcal{N}_J$ form a $1/4$-net for $\mathbb{S}^p(k)$. Define $u_J \in \text{argmax}_{u \in B_J} \|Xu\|^2$

and let $v_J$ be an element in $\mathcal{N}_J$ closest in Euclidean distance to $u_J$. Define $A := X^\top X - I_p$. We have

$$|u_J^\top A u_J| \leq |v_J^\top A v_J| + |(u_J - v_J)^\top A v_J| + |u_J^\top A (u_J - v_J)| \leq \max_{u \in \mathcal{N}_I} |u^\top A u| + \frac{1}{2} \sup_{u \in \mathbb{S}^p(k)} |u^\top A u|.$$

Hence

$$\sup_{u \in \mathbb{S}^p(k)} |u^\top A u| \leq 2 \max_{u \in \mathcal{N}} |u^\top A u|,$$

which verifies the claim. $\qquad\square$

*Proof of Proposition 2.* By definition, $\|X^\top X - I_p\|_{\mathrm{op},k} \leq \theta$ is equivalent to $X \in \mathrm{RIP}_{n,p}(k,\theta)$. Moreover, by Proposition 1, $X \in \mathrm{RIP}_{n,p}(k,\theta)$ with probability converging to 1, under $\tilde{Q}^{\otimes(n \times p)}$. The certifier hence satisfies the two desired properties. $\qquad\square$

*Proof of Proposition 3.* The proposed certifier is clearly polynomial time computable (it has time complexity $O(np^2)$). To verify that it is a certifier, we check that (i) $\psi_n^{-1}(1) \subseteq \mathrm{RIP}_{n,p}(k,\theta)$ and (ii) $\liminf_{n\to\infty} \tilde{Q}^{\otimes(n \times p)}(\psi_n^{-1}(1)) > 2/3$.

For (i), on the event $\|X^\top X - I_p\|_\infty \leq 14\sigma^2 \sqrt{\frac{\log p}{n}}$, for any index set $T \in \{1, \ldots, p\}$ of cardinality $k$, we have $\|X_{*T}^\top X_{*T} - I_k\|_\infty \leq 14\sigma^2 \sqrt{\frac{\log p}{n}}$, which implies that

$$\|X_{*T}^\top X_{*T} - I_k\|_{\mathrm{op}} \leq 14\sigma^2 k \sqrt{\frac{\log p}{n}} \leq \theta$$

For (ii), let $Y_n \sim \chi_n^2$. Using Lemma 5 and the fact that for any $A \in \mathbb{R}^{p \times p}$

$$\|A\|_\infty = \sup_{S \subseteq \{1,\ldots,p\}, \#S=2} \|A_{SS}\|_\infty \leq \sup_{S \subseteq \{1,\ldots,p\}, \#S=2} \|A_{SS}\|_{\mathrm{op}} = \|A\|_{\mathrm{op},2}$$

we get

$$\mathbb{P}\left\{\|X^\top X - I_p\|_\infty \leq 14\sigma^2 \sqrt{\frac{\log p}{n}}\right\} \geq \mathbb{P}\left\{\sup_{u \in \mathbb{S}^p(2)} \big|\|Xu\|_2^2 - 1\big| \leq 14\sigma^2 \sqrt{\frac{\log p}{n}}\right\}$$

$$\geq 1 - 2\binom{p}{2} 9^2 \exp\left\{-\frac{n}{256\sigma^4} \frac{196\sigma^4 \log p}{n}\right\}$$

$$\geq 1 - 81p^2 \exp\{-3\log p/4\} \to 1.$$

as desired. $\qquad\square$

**Lemma 2.** *Let $Z$ be a non-negative random variable and $r \geq 2$, then*

$$\mathbb{E}(Z^r) \geq \mathbb{E}(|Z - \mathbb{E}Z|^r).$$

*In other words, centring a nonnegative random variable shrinks its second or higher absolute moments.*

*Proof.* Let $\mu := \mathbb{E}(Z)$ and define $Y = Z - \mu$. Let $P$ denote the probability measure on $\mathbb{R}$ associated with random variable $Y$. Hence $\int_{[-\mu,\infty)} y\, dP(y) = 0$. Without loss of generality, we may assume that $Z$ is not a point mass. Then $\int_{[-\mu,0]} (-y)\, dP(y) = \int_{(0,\infty)} y\, dP(y) = A$ for some $A > 0$. For any measureable function $f : \mathbb{R} \to [0,\infty)$, we may write

$$A\int_{[-\mu,\infty)} f(y)\, dP(y) = \int_{[-\mu,0]} (-v)\, dP(v) \int_{(0,\infty)} f(u)\, dP(u) + \int_{(0,\infty)} u\, dP(u) \int_{[-\mu,0]} f(v) dP(v)$$

$$= \int_{u \in (0,\infty)} \int_{v \in [-\mu,0]} \left(\frac{u}{u-v} f(v) - \frac{v}{u-v} f(u)\right)(u - v)\, dP(v)\, dP(u).$$

$$(5)$$

Let $(U, V)$ be a bivariate random vector having probability measure

$$\frac{1}{A}(u - v)\mathbb{1}_{(0,\infty)}(u)\mathbb{1}_{[-\mu,0]}(v)\, dP(u)\, dP(v)$$

on $\mathbb{R}^2$ (that this is a probability measure follows from substituting $f(y) \equiv 1$ in (5)). Then (5) can be rewritten as

$$\mathbb{E}\{f(Y)\} = \mathbb{E}\left\{\frac{U}{U - V}f(V) - \frac{V}{U - V}f(U)\right\}.$$

Now consider choosing $f$ to be $f_1(y) = |y|^r$ and $f_2(y) = (y + \mu)^r$ respectively in the above equation. Note that for $u \in (0, \infty)$ and $v \in [-\mu, 0]$ and $r \geq 2$, we always have

$$uf_2(v) - vf_2(u) \geq -vf_2(u) \geq -v(u - v)^r \geq (-v)^r u + (-v)u^r \geq uf_1(v) - vf_1(u).$$

Therefore,

$$\begin{aligned}
\mathbb{E}(|Y|^m) &= \mathbb{E}\left\{\frac{U}{U - V}f_1(V) - \frac{V}{U - V}f_1(U)\right\} \\
&\leq \mathbb{E}\left\{\frac{U}{U - V}f_2(V) - \frac{V}{U - V}f_2(U)\right\} = \mathbb{E}(|Y + \mu|^m),
\end{aligned}$$

as desired. $\qquad\square$

**Lemma 3.** *Suppose $X$ is a sub-Gaussian random variable with parameter $\sigma$ and median $\xi$. Let $X^+ = X \mid X \geq \xi$ and $X^- = X \mid X < \xi$. Then $X^+ - \mathbb{E}X^+$ and $X^- - \mathbb{E}X^-$ are both sub-Gaussian with parameters are most $c\sigma$ for some absolute constant $c$.*

*Proof.* By Vershynin (2012, Lemma 5.5), $X$ is sub-Gaussian with parameter $\sigma$ implies that $(\mathbb{E}|X|^p)^{1/p} \leq c_1\sigma\sqrt{p}$ for some absolute constant $c_1$. Hence by Lemma 2, we have

$$\mathbb{E}(|X^+ - \mathbb{E}X^+|^p)^{1/p} \leq (\mathbb{E}|X^+|^p)^{1/p} = 2(\mathbb{E}|X\mathbb{1}\{X \geq \xi\}|^p)^{1/p} \leq 2c_1\sigma\sqrt{p}.$$

Using Vershynin (2012, Lemma 5.5) again, we have that $X^+ - \mathbb{E}X^+$ is sub-Gaussian with parameter at most $c\sigma$ for some absolute constant $c$. A similar argument holds for $X^- - \mathbb{E}X^-$. $\qquad\square$

**Lemma 4.** *Suppose $X$ is a random variable satisfying $\mathbb{E}e^{\lambda X} \leq e^{\sigma^2\lambda^2/2}$ for all $\lambda \in \mathbb{R}$. Define $Y = X^2 - \mathbb{E}X^2$. Then $\mathbb{E}e^{\lambda Y} \leq e^{16\sigma^4\lambda^2}$ for all $|\lambda| \leq \frac{1}{4\sigma^2}$.*

*Proof.* By Markov's inequality,

$$\mathbb{P}(|X| \geq t) = \mathbb{P}(X \geq t) + \mathbb{P}(-X \geq t) \leq e^{-t^2/\sigma^2}\mathbb{E}(e^{tX/\sigma^2}) + e^{-t^2/\sigma^2}\mathbb{E}(e^{-tX/\sigma^2}) \leq 2e^{-t^2/(2\sigma^2)}.$$

From Lemma 2, for $r \geq 2$

$$\mathbb{E}(|Y|^r) \leq \mathbb{E}(|X|^{2r}) = \int_0^\infty \mathbb{P}(|X| \geq t)(2r)t^{2r-1}\, dt \leq \int_0^\infty 4rt^{2r-1}e^{-t^2/(2\sigma^2)}\, dt = 2(2\sigma^2)^r\Gamma(r+1).$$

Consequently, if $|2\sigma^2\lambda| \leq 1/2$, then

$$\mathbb{E}e^{\lambda Y} = \sum_{r=0}^\infty \frac{\lambda^r\mathbb{E}Y^r}{r!} \leq 1 + 2\sum_{r=2}^\infty (2\sigma^2\lambda)^r \leq 1 + 16\sigma^4\lambda^2 \leq e^{16\sigma^4\lambda^2},$$

as desired. $\qquad\square$

**Lemma 5.** *Let $X_1, X_2, \ldots, X_n$ be independent sub-Gaussian random variables with sub-Gaussian parameters at most $\sigma$. Let $Y_i := X_i^2 - \mathbb{E}X_i^2$. Then*

$$\mathbb{P}\left(\sum_{i=1}^n Y_i \geq \theta\right) \leq \exp\left\{-\left(\frac{\theta^2}{64n\sigma^4} \wedge \frac{\theta}{8\sigma^2}\right)\right\}$$

$$\mathbb{P}\left(\sum_{i=1}^n Y_i \leq -\theta\right) \leq \exp\left\{-\frac{\theta^2}{64n\sigma^4}\right\}$$

*Proof.* Using Markov's inequality, we have

$$\mathbb{P}\left(\sum_{i=1}^{n} Y_i \geq \theta\right) = \mathbb{P}\left(e^{\lambda \sum_i Y_i} \geq e^{\lambda \theta}\right) \leq e^{-\lambda \theta} \prod_i \mathbb{E} e^{\lambda Y_i}.$$

Set $\lambda = \frac{\theta}{32n\sigma^4} \wedge \frac{1}{4\sigma^2}$. By Lemma 4, we have

$$\mathbb{P}\left(\sum_{i=1}^{n} Y_i \geq \theta\right) \leq e^{-\lambda\theta + 16\lambda^2 n\sigma^4} \leq e^{-\lambda\theta/2},$$

which establishes the first desired inequality. Applying the same argument with $-Y_i$ in place of $Y_i$ we get

$$\mathbb{P}\left(\sum_{i=1}^{n} Y_i \leq -\theta\right) \leq \exp\left\{-\left(\frac{\theta^2}{64n\sigma^4} \wedge \frac{\theta}{8\sigma^2}\right)\right\}. \tag{6}$$

Taylor expand the moment generating function of $X_i$ around 0, we have $\mathbb{E} X_i^2 \leq \sigma^2$. Hence we may assume $\theta \leq n\sigma^2$. Then we have

$$\frac{\theta^2}{64n\sigma^4} < \frac{\theta}{8\sigma^2},$$

which together with (6) implies the desired result. $\qquad\square$

**Lemma 6.** *Suppose $n\ell$ balls are arranged in an array of $n$ rows and $\ell$ columns and $k$ balls $(k < n)$ are chosen uniformly at random. Let $V_i$ be the number of chosen balls in row $i$ and $V = (V_1, \ldots, V_n)^\top$. Then*

$$\mathbb{P}\left(\|V\|_0 \leq k - \frac{k^2}{2n} - \sqrt{k \log k}\right) \leq \frac{1}{k^2}.$$

*Moreover, if $k \leq n^\gamma$ for some $\gamma < 1$, then*

$$\mathbb{P}\left(\|V\|_\infty \geq a\right) \leq n^{1-a(1-\gamma)}\left(1 - n^{-(1-\gamma)}\right).$$

*Proof.* Let $U_i$ be the number of balls chosen in row $i$ when balls are drawn with replacement from the array and $U = (U_1, \ldots, U_n)^\top$. Then $\|V\|_0$ is stochastically larger than $\|U\|_0$ and $\|V\|_\infty$ is stochastically smaller than $\|U\|_\infty$. So it suffices to show the desired inequalities with $U$ replacing $V$. In the following argument, we consider only drawing with replacement.

Let $\mathcal{X} = \{e_1, \ldots, e_n\}$ where $e_i$ denotes the $i$th standard basis vector in $\mathbb{R}^n$. For $1 \leq r \leq k$, let $X_r$ be uniformly distributed in $\mathcal{X}$. Then $U \stackrel{d}{=} \sum_{r=1}^{k} X_r$. We note that changing the value of any of the $X_r$ affects the value of $\|U\|_0$ by at most 1. By McDiarmid's inequality (McDiarmid, 1989), we have that for any $t > 0$,

$$\mathbb{P}\left(\|U\|_0 - \mathbb{E}\|U\|_0 \leq -t\right) \leq e^{-\frac{2t^2}{k}}. \tag{7}$$

For $1 \leq i \leq n$. Define $J_i = \mathbb{1}\{\text{no ball is chosen in row } i\}$, then

$$\mathbb{E}\|U\|_0 = n - \sum_{i=1}^{n} \mathbb{E} J_i = n - n(1 - 1/n)^k \geq k\left(1 - \frac{k}{2n}\right).$$

Thus, together with (7), we have

$$\mathbb{P}\left(\|U\|_0 \leq k - \frac{k^2}{2n} - \sqrt{k \log k}\right) \leq \mathbb{P}\left(\|U\|_0 - \mathbb{E}\|U\|_0 \leq -\sqrt{k \log k}\right) \leq e^{-2\log k} = k^{-2},$$

as desired. For the second inequality,

we have by union bound that

$$\mathbb{P}(\|U\|_\infty \geq a) \leq n\mathbb{P}(U_1 \geq a) = n\sum_{s=a}^{k} \binom{k}{s} n^{-s}$$

$$\leq n\sum_{s=a}^{\infty} (k/n)^s = n\frac{(k/n)^a}{1 - k/n} \leq n^{1-a(1-\gamma)}\left(1 - n^{-(1-\gamma)}\right)^{-1},$$

as desired. $\qquad\square$