[Reviews · NeurIPS 2016]

Reviewer 1

Summary

The paper studies average-case certifiers for checking that a design matrix verifies the restricted isoperimetry property (RIP). An average-case certifier never wrongly declares that a matrix is RIP, and the authors prove that no computationally efficient average-case certifier can be obtained in certain regimes. The scheme of proof relies on the Planted Clique hypothesis.

Qualitative Assessment

The paper is pleasant to read and contains notable contributions. The introduction clearly states the paper's ambitions, even for non-expert readers. The main result combined with Figure 1 states what computational tractability for average-case certifiers a practitioner could hope for in different regimes. In my opinion, this is a fine piece of work.

Confidence in this Review

1-Less confident (might not have understood significant parts)


Reviewer 2

Summary

This paper focuses on testing whether a given matrix satisfies a restricted isometry property. This is known to be a hard problem. The authors identify a regime for the parameters in which there exists a polynomial time algorithm that is conservative, i.e., never assigns the RIP property to a matrix that does not satisfy it, and rejects only a small proportion of matrices that do satisfy the property. In addition, by making a connexion with random graph theory, they show that there is no polynomial time algorithm in any larger regime.

Qualitative Assessment

This work is very well presented and provide important results for certifying the RIP property. I am only concerned with the negative result given in Theorem 4. It relies on Assumption (A1), which I believe is not an assumption but a conjecture, which is not guaranteed to be true. Overall, I am favorable for acceptance of this work at NIPS.

Confidence in this Review

2-Confident (read it all; understood it all reasonably well)


Reviewer 3

Summary

A matrix of size (n,p), whose coefficients are independently drawn from suitable random distributions, satisfies with high probability the RIP property with parameters k,t, provided that n >> k log p / (t*t). The question considered in this article is: is there a polynomial-time algorithm that, given such a random matrix, is able to certify, in at least 2/3 of the cases, that this matrix indeed satisfies RIP? The answer is known to be positive when n is of the order of (k*k) log p / (t*t). This article shows that, for any alpha < 1, there is a sequence of (n,k,t) going to infinity, such that n >> k^(1+alpha) / (t*t), for which no certifier exists. The result holds under the assumption that a problem of dense subgraph testing can also not be solved with relatively large probability in polynomial time, in a certain regime.

Qualitative Assessment

I think that the result of this article is interesting and elegant. I would be happy to see it published. However, the proofs, in the appendix, contain a number of typos that need to be corrected. There is also an inequality that I do not understand, at the end of the proof of the main theorem; I would appreciate additional explanations about it. Regarding the presentation, I find the article well-written. I have two regrets, but they are relatively minor: - I think it would be better to underline before the very end of the article that the proof does not currently work in the regime n >> k^(1+alpha) and theta constant. - I would have liked the authors to give a general picture of the proof in the main part of the article, more detailed that the paragraph that follows Theorem 4. Typos and remarks: - Line 16: "ee" -> "see". - Theorem (informal): I think the statement would be more clear with "for any alpha" at the beginning of the sentence. - Line 80: "an a" -> "a". - "measureable" -> "measurable" (appears twice). - Line 178: "lim" -> "lim inf". - Line 256: I think A_G has not been defined. - Lines 284/285, "shows through a reduction to the planted dense subgraph problem": this part of the sentence is not clear. - In the appendix, equation after line 9: "\approx" -> "<<" ? - Algorithm 1: I think the definition of k should appear before the one of l. - Line 14: the coefficients of X to not exactly have distribution \tilde Q, because of the averaging. - Equation after line 14: it does not necessarily go to zero. I think we only know that the lim sup is at most 1/3. - Line 15: the statement holds only "with high probability". - Equation after line 16: I do not think that the second part is necessarily true. You have to specify how you choose theta. - Equation after line 18: the lim inf must actually be zero. - Equation after line 23 should be broken in two lines. - Equation after line 34 is only valid for epsilon under some constant value. - Equation (3): I do not understand where the k comes from in the denominator, inside the exponential. - Line 46: shouldn't v be padded with zeros, so as to be of size p? - Equation after line 49: I think that the zero-norm of \tilde v should be replaced with the cardinality of \tilde S. - Equation after line 49 (this is the one that I do not understand): from what I understand, the authors use as a lower bound for the 1-norm the sum of Z^(a,b) v^(b). Is my understanding correct? Because then, aren't there cross terms Z^(a,b) v^(b')? - Line 53: I think the authors could give a slightly more detailed justification, because \tilde v and \tilde X are not independent. - Line 57: I think a l^2 is missing in the denominator. - Statement of Lemma 1: I think the three inequalities are in the wrong direction. I also have the impression that the right hand sides should be squared (because of the application of Chebyshev's inequality). - Equation after line 67: in the first line, the 'v' should be a 'u'. At the end, there should be only one dot. - Equation after line 78: the inequality should be reversed. - Equation after line 90: I think that there should not be a square in the right hand side. I also do not think that the second and third terms of the middle part can be directly majorized by 1/2 |u_J^TAu_J|. - Line 97: "O(n^2p)" -> "O(np^2)"? - Line 98: "lim" -> "lim inf". - Line 101: "and fact that" -> "and the fact that". - Equation after line 102: shouldn't the term in the exponential be divided by 4, as in the equation after line 85, where the denominator is 256, and not 64? - Line 107: "R" -> "mathbb{R}". - Lines 114 to 116: is b equal to mu? - Equation after line 127: I think that the second equality is an inequality. - Statement of Lemma 6, and last equation of the proof: I think (1-n^(1-gamma)) is (1-n^(1-gamma))^(-1). - Line 146: "smaller than ||U||_0" -> "smaller than ||U||_\infty". - Line 149: "any one" -> "any of the".

Confidence in this Review

2-Confident (read it all; understood it all reasonably well)


Reviewer 4

Summary

The paper looks at the question of certifying restricted isometric property for matrix X. Random matrices typically satisfy this, but the goal is to show that this is indeed the case. The worst-case problem is NP-hard. This paper looks at average case instance and shows that it is also likely to be hard in a certain range of parameters.

Qualitative Assessment

The paper is written quite well. One weakness is that $\theta_n$ -> 0 as n -> \infty. One usually cares about constant $\theta$. So the hardness result may be less interesting for practical problems. However, it may be of interest to part of the NIPS community and may lead to further progress. The techniques used to prove the lower bound are by now standard (following Berthet and Rigollet (2013), and then several subsequent papers of this kind).

Confidence in this Review

2-Confident (read it all; understood it all reasonably well)


Reviewer 5

Summary

The authors provides an average-case certifier for determining if a matrix is RIP, and a Las Vegas algorithm to construct RIP matrices. Based on the planted dense subgraph certification assumption, the authors also show that there exists a regime where no computational efficient procedures can determine if a matrix is RIP.

Qualitative Assessment

This paper seems to follow a line of previous papers on computational and information theoretic gaps. I believe it's a technically solid paper, but RIP certification may not be of much interest to the general NIPS community.

Confidence in this Review

2-Confident (read it all; understood it all reasonably well)


Reviewer 6

Summary

The paper consider the problem of testing whether a given matrix satisfies the restricted isometry property (RIP), which is a common assumption used in algorithms for compressive sensing and, to some extent, sparse linear regression. In contrast to prior work that considered only worst-case hardness, the current paper shows an average-case hardness result that applies to iid sub-Gaussian matrices, which is an important class of measurement matrices. The authors establish a conditional hardness result for this testing problem via reduction to the planted dense subgraph problem, which is a weaker variant of the planted clique problem that is commonly used in average-case hardness results for statistical problems.

Qualitative Assessment

The average-case hardness setting considered in the current work is significantly more important than the worst-case hardness results for RIP certification in prior work because the latter usually deals with somewhat artificial instances. The paper is also very well written. As pointed out by the authors, a limitation of their approach is that it only applies to RIP-constants theta that scale with n. This is an important restriction because several compressive sensing algorithms are known that work for sufficiently small but constant theta. Hence it could still be the case that certifying RIP in this limited regime is easier. (As a related writing comment: it would be helpful if the authors gave the dependence of theta on n in the related discussion near the bottom on page 7). Another question is how important RIP certification is in practice. It is certainly true that an efficient RIP-certifier would be a convenient tool to have. However, in compressive sensing, the measurement ensembles (subsampled Fourier, etc.) have guarantees with high probability that are sufficient for engineering applications. In sparse linear regression settings, the common assumption seems to be that RIP is too restrictive and less stringent assumptions on the design matrix are more relevant. For instance, certifying that a matrix does not satisfy RIP is easy if there are two very correlated features. Maybe the authors can comment on this point in the paper and / or rebuttal. Nevertheless, I find this paper a good submission to NIPS and strongly recommend accepting it.

Confidence in this Review

3-Expert (read the paper in detail, know the area, quite certain of my opinion)